# High-Fidelity ANN-to-SNN Conversion
# via Closed-Loop CKA Distillation

**Bozhou Li** [1]  **Chubo Liu** [1 2]  **Yan Ding** [1]  **Yufeng Zhang** [1]  **Zhuo Tang** [1]  **Kenli Li** [1]

## Abstract

ANN-to-SNN conversion offers energy-efficient inference but faces a fidelity-latency trade-off due to open-loop error accumulation. While conversion-aware training mitigates this, it sacrifices the generality of using off-the-shelf ANNs. We propose a closed-loop fine-tuning framework that calibrates these errors without altering the source model. Our approach employs a Dual Alignment Mechanism, utilizing global Kullback-Leibler divergence for output distillation and introducing an adaptive local Centered Kernel Alignment constraint, weighted by initial conversion loss, for feature alignment. We uncover a critical time-dependent dynamic: local constraints are essential for stabilizing representations in low-latency regimes (e.g., $T = 8$) where global gradients are unstable, whereas global alignment drives fidelity at higher time steps. Experiments on CIFAR-10 demonstrate that our method achieves over 99% of source ANN accuracy at $T = 32$ (e.g., ResNet-18: 96.38% vs. 96.39%). Furthermore, this fine-tuning acts as a regularizer, yielding SNNs with input noise robustness that matches or exceeds the source ANN.

## 1. Introduction

Spiking Neural Networks (SNNs), the third generation of neural models, process information via discrete, binary spike events over time (Maass, 1997; Tavanaei et al., 2019). Unlike traditional Artificial Neural Networks (ANNs), this event-driven paradigm avoids dense multiply-accumulate operations, offering inherent energy efficiency, especially

[1]Hunan University, Changsha, China [2]The Ministry of Education Key Laboratory of "Fusion Computing of Supercomputing and Artificial Intelligence", Changsha, China. Correspondence to: Chubo Liu <liuchubo@hnu.edu.cn>, Kenli Li <likenli@hnu.edu.cn>.

*Proceedings of the $43^{rd}$ International Conference on Machine Learning*, Seoul, South Korea. PMLR 306, 2026. Copyright 2026 by the author(s).

*Table 1.* Time steps requirements and relative latency for target fidelity levels.

| Model | Peak Acc. | T (95%) | Relative Latency |
|---|---|---|---|
| Rs18-C10 | 94.31 | 42 | 1.00x |
| Rs18-C100 | 71.70 | 122 | 2.90x |
| Rs50-C10 | 94.24 | 200 | 4.76x |
| Rs50-C100 | 79.17 | 416 | 9.90x |

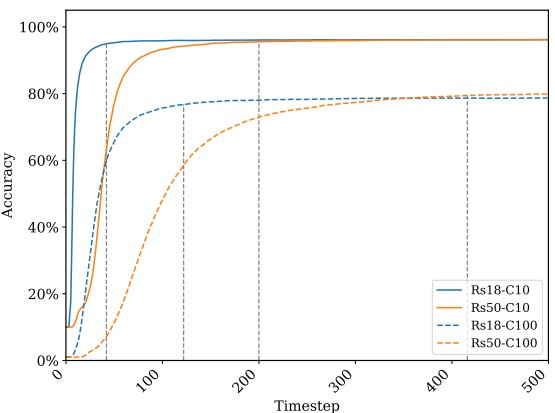

*Figure 1.* Accuracy convergence curves versus time steps ($T$). Dashed vertical lines indicate the $T(95\%)$ convergence points.

on neuromorphic hardware (Merolla et al., 2014; Akopyan et al., 2015; Ma et al., 2017; Davies et al., 2018; Roy et al., 2019; Pei et al., 2019).

However, training SNN to match ANN performance is challenging due to the non-differentiable spike generation process. While several training paradigms exist, ANN-to-SNN conversion (ANN2SNN) (Diehl et al., 2015) offers a compelling compromise: it supports deep networks with high training efficiency by leveraging pre-trained ANNs. Despite its promise, the fidelity and latency of this conversion process remain key bottlenecks.

To motivate our approach, we first examine how model complexity and task difficulty affect the time-step requirements in conventional ANN-to-SNN conversion using the SpikingJelly framework (Fang et al., 2023). We define $T(95\%)$ as

*Table 2.* Comparison of three major SNN training and conversion paradigms.

| Aspect | | STDP | STBP | ANN2SNN |
|---|---|---|---|---|
| **Core Principle** | Target | Biologically plausible | Direct Optimization | Knowledge Transfer |
| | Learning Paradigm | Unsupervised | Supervised | Indirectly Supervised |
| | Learning Rule | Hebb-based plasticity | Surrogate Gradient Descent | Weight inheritance |
| **Performance** | SNN Accuracy | Low | Very High | High |
| | Network Depth | Shallow | Deep | Very Deep |
| | Training Efficiency | Medium | Low | Very High |
| **Limitations** | | ① ③ ④ | ② ③ ④ | ④ ⑤ |
| **Remarks:** | ① Limited to simple tasks. | ② High dataset dependency. | ③ Retrain from scratch. | |
| | ④ High inference latency. | ⑤ Not inherently event-driven. | | |

the time steps needed to reach 95% of the final accuracy (at $T = 1000$). Fig. 1 shows the accuracy convergence behavior of ResNet-18 and ResNet-50 on CIFAR-10 and CIFAR-100 across different time steps, and Table 1 quantifies the latency cost. Here, Relative Latency is normalized by the Rs18-C10 baseline. The results reveal a critical challenge: deeper networks and more complex tasks require substantially larger $T$ to achieve high-fidelity conversion. For instance, while ResNet-18 on CIFAR-10 reaches this target at $T = 42$, ResNet-50 on CIFAR-100 requires $T = 416$. This represents a dramatic $9.90\times$ increase in temporal burden, which compromises the SNN latency and energy advantages and highlights the pressing need for effective optimization strategies.

The fidelity of this conversion process is predominantly hindered by its open-loop nature, where quantization errors from discrete spike events accumulate and propagate forward. In deep architectures, this cascading error effect significantly distorts model feature representations, leading to severe performance degradation (Diehl et al., 2015; Rueckauer et al., 2017; Sengupta et al., 2019; Han et al., 2020). While several works have attempted to mitigate this issue, they often compromise the conversion's generality by requiring methods that alter the source ANN's training process or activation functions (Rathi & Roy, 2020; Deng & Gu, 2021; Ding et al., 2021; Bu et al., 2023; Jiang et al., 2023; Ho & Chang, 2023). This sacrifices the key advantage of using off-the-shelf, pre-trained models. To address this, we propose a Closed-Loop Fine-Tuning Framework that combines Spatio-Temporal Backpropagation (STBP) (Wu et al., 2018; Neftci et al., 2019) and ANN-SNN conversion approaches. Our framework is capable of correcting feature representation errors after the initial conversion without modifying the source ANN.

Our framework must restore fidelity at both the output and intermediate layers. While global output matching is straightforward, correcting internal, layer-wise feature drift requires a robust metric for representational similarity. We employ Centered Kernel Alignment (CKA) (Kornblith

et al., 2019; Saha et al., 2022), a similarity index that is invariant to isotropic scaling. Unlike prior linear methods, CKA provides a stable comparison of representations across networks, making it ideal for minimizing the feature-space gap between ANNs and SNNs (Saha et al., 2022; Li et al., 2023).

Our main contributions are summarized as follows:

- We propose a closed-loop fine-tuning method that, by incorporating STBP, significantly improves ANN-to-SNN conversion fidelity without altering the source ANN, ensuring broad applicability.

- We introduce a dual alignment framework, combining global output feedback and local feature alignment, to synergistically correct for layer-wise error accumulation.

- Extensive experiments demonstrate that our method substantially reduces the accuracy gap, shortens the required inference time for high performance, and enhances SNN robustness against noisy inputs, outperforming existing conversion techniques.

## 2. Related Work

As summarized in Table 2, current SNN training methods are primarily categorized into three paradigms: Spike-Timing-Dependent Plasticity (STDP), Spatio-Temporal Backpropagation (STBP), and ANN-to-SNN conversion (ANN2SNN). These paradigms present a clear trade-off. STDP (Markram et al., 2011) is biologically plausible and training-efficient but is restricted to shallow, low-accuracy networks. STBP (Wu et al., 2018; Neftci et al., 2019) achieves high accuracy by training SNNs directly with surrogate gradients, but it suffers from significant computational overhead. The third paradigm, ANN2SNN, which obtains SNN models by copying pre-trained ANN weights, has become a mainstream approach. However, its conversion fidelity and inference latency remain key challenges (Bu et al., 2025), which form the primary focus of this work.

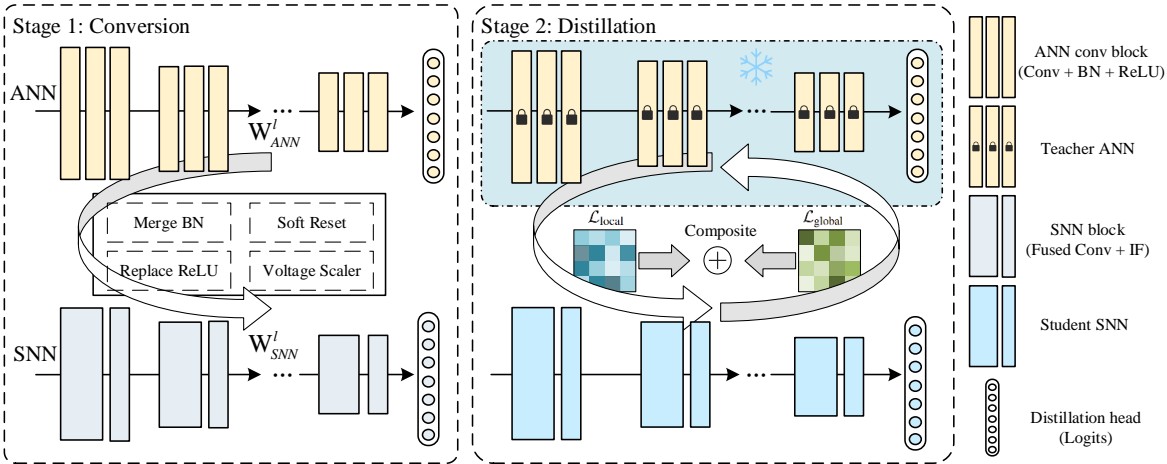

*Figure 2.* The proposed Closed-Loop Fine-Tuning Framework, featuring Stage 1: ANN-to-SNN Conversion and Stage 2: Distillation with Dual Alignment.

## 2.1. Direct SNN Training Methods

Direct training methods build SNNs from scratch using spike-based learning rules. The first approach, Spike-Timing-Dependent Plasticity (STDP), is an unsupervised, biologically plausible learning rule based on Hebbian principles (Bi & Poo, 1998; Markram et al., 2011). While computationally efficient and suitable for online learning, STDP-based methods are typically limited to shallow networks and simpler tasks.

The second approach, Spatio-Temporal Backpropagation (STBP), overcomes the performance limitations of STDP by enabling direct, end-to-end supervised training. The primary challenge of this optimization, the non-differentiable spike generation, was overcome by the Surrogate Gradient (SG) method (Wu et al., 2018; Neftci et al., 2019). The core idea is to approximate the derivative of the hard-threshold spiking function with a smooth, continuous surrogate during the backward pass. This line of research has produced highly competitive models. For instance, SEW ResNet (Fang et al., 2021) introduced a deep residual learning framework to address gradient vanishing issues. Further advancements have boosted performance by improving temporal sequence learning (Zhang & Li, 2020), enabling deeper architectures (Zheng et al., 2021), developing efficient input encoding schemes (Rathi & Roy, 2020), and refining the gradient approximation itself (Wu et al., 2019; Li et al., 2021b).

## 2.2. ANN-to-SNN Conversion

This paradigm inherits parameters from an isomorphic ANN to avoid expensive training. Early conversion techniques focused on mitigating error by establishing that SNN firing rates can approximate ANN activations, replacing the

ANN's ReLU function with spiking neurons, and applying weight and threshold balancing for shallow networks (Cao et al., 2015; Diehl et al., 2015). Subsequent efforts scaled conversion to deeper architectures. Key conceptual advancements include data-driven normalization to set firing thresholds (Rueckauer et al., 2017), methods for absorbing batch normalization parameters into convolutional weights (Sengupta et al., 2019), and soft reset mechanisms to reduce information loss (Han et al., 2020). These techniques collectively enabled the successful conversion of deep architectures like VGG and ResNet.

Despite this progress, conversion fidelity remains the central challenge. The process is fundamentally lossy, as it approximates continuous ANN activations with discrete spike events. Theoretical analyses identify primary error sources stemming from this approximation, including quantization error, clipping error, and temporal dynamics mismatch (Han et al., 2020; Deng & Gu, 2021; Li et al., 2021a; Meng et al., 2022; Bu et al., 2023).

One influential approach to mitigate these errors is conversion-aware training. In this strategy, the source ANN is modified to be more amenable to conversion. A state-of-the-art method, QCFS (Bu et al., 2023), replaces the ANN's ReLU with a parameterized function designed to mimic the SNN's discrete firing rate, pre-quantizing activations to minimize the mapping error. Similarly, recent works have integrated quantization-aware frameworks (Gao et al., 2023; Hu et al., 2023) to bridge the ANN-SNN gap. Other methods using this strategy include the Rate Norm Layer (Ding et al., 2021) or theoretical optimizations like bias shifting (Deng & Gu, 2021).

A distinct strategy focuses on post-conversion optimization, which adjusts SNN parameters after the initial conversion.

Early work in this area proposed static, data-driven layer-wise parameter correction to reduce errors (Li et al., 2021a). This concept was extended in frameworks compensating for residual potentials (Wang et al., 2023) and offset spikes (Hao et al., 2023), or correcting forward temporal bias to enhance low-latency fidelity (Wu et al., 2024). However, these open-loop adjustments can cause residual errors to accumulate.

This limitation necessitates a closed-loop optimization process. This second paradigm of post-conversion optimization, and the focus of this work, is post-conversion fine-tuning. In this approach, STBP is applied to an already-converted SNN to correct residual errors. A prominent strategy within this paradigm is employing Knowledge Distillation (KD) (Hinton et al., 2015) to fine-tune the converted SNN. This often involves enforcing consistency between the teacher ANN and student SNN outputs, typically by minimizing the Kullback-Leibler divergence between their respective softmax distributions (Yu et al., 2025). Hybrid strategies involving both ANN and SNN computations have also been explored to stabilize optimization (Liu et al., 2024). A more granular approach involves aligning intermediate feature representations, for which CKA (Kornblith et al., 2019) has emerged as a robust metric (Saha et al., 2022; Zhou et al., 2024).

## 3. Methodology

### 3.1. Closed-Loop Framework Overview

As illustrated in Fig. 2, the proposed Closed-Loop Fine-Tuning Framework operates in two distinct stages. Stage 1: ANN2SNN Conversion maps the weights of the pre-trained source ANN to an SNN of the same architecture. This conversion involves four key steps: merging the Batch Normalization layers into the preceding Convolution layer weights; mitigating information loss after spiking via a soft reset mechanism; replacing the ReLU activation function with the Integrate-and-Fire (IF) spiking neuron model; and applying a Voltage Scaler to normalize the weights $W_{\text{ANN}}^l$ to the SNN domain $W_{\text{SNN}}^l$. Despite inheriting the weights, the resulting SNN suffers from performance degradation due to uncorrected conversion errors, thus necessitating the second stage.

Stage 2: Distillation addresses the layer-wise accumulated conversion errors by fine-tuning the Student SNN using a knowledge distillation approach. The Teacher ANN, with its weights frozen, guides the trainable Student SNN through the closed-loop fine-tuning framework. This stage employs a composite loss that achieves dual alignment via a global output distillation signal ($\mathcal{L}_{\text{global}}$) and an adaptive local feature constraint ($\mathcal{L}_{\text{local}}$). $\mathcal{L}_{\text{global}}$ aligns the SNN's firing rate logits with the Teacher's logits, while $\mathcal{L}_{\text{local}}$, derived from CKA, suppresses layer-wise drift by enforcing representa-

tational similarity at intermediate stages. This systematic correction updates only the SNN weights, thereby effectively restoring the fidelity of the converted model.

### 3.2. ANN-to-SNN Conversion

The first stage of our framework converts a pre-trained ANN into a functionally equivalent SNN by preserving its learned parameters while adapting the architecture for spiking computation. To optimize the SNN for inference, Batch Normalization (BN) layers are first merged into their preceding convolutional layers (Sengupta et al., 2019). The BN parameters are folded directly into the weights and biases, creating a simplified architecture that performs an identical linear transformation without requiring separate BN operations during simulation. All ReLU activation functions are then replaced with IF spiking neurons. This step transforms the network from a static, continuous-valued system to a dynamic, event-driven one.

To mitigate information loss from hard resets, a soft-reset mechanism is implemented (Han et al., 2020). This soft-reset principle reduces a neuron's membrane potential by its firing threshold upon firing, rather than resetting to zero. This allows residual potential, or sub-threshold information, to be preserved and integrated into future time steps, which is critical for deep networks.

A voltage scaler strategy is employed for threshold balancing to address the potential performance degradation from activation-threshold mismatch. Following a data-driven calibration approach (Rueckauer et al., 2017; Sengupta et al., 2019), we run inference on a calibration set to collect activation statistics. The firing threshold $V_{th}$ for each layer's neurons is then set to the 99.9th percentile of the recorded ANN activations for that layer. This robust scaling prevents rare, high-magnitude activations from skewing the threshold, ensuring a stable and efficient firing rate.

### 3.3. Dual Alignment Loss Function

The Student SNN's optimization is guided by a composite loss function that balances task performance, global output alignment, and local feature preservation. The total loss is described by

$$\mathcal{L}_{\text{total}} = (1 - \alpha)\mathcal{L}_{\text{task}} + \alpha\left(\beta\mathcal{L}_{\text{global}} + (1 - \beta)\mathcal{L}_{\text{local}}\right), \quad (1)$$

where $\mathcal{L}_{\text{task}}$, $\mathcal{L}_{\text{global}}$, and $\mathcal{L}_{\text{local}}$ are the loss components, and $\alpha, \beta \in [0, 1]$ are the key hyperparameters that control the balance of these objectives.

**Task Loss ($\mathcal{L}_{\text{task}}$)** is the standard cross-entropy (CE) loss, which measures the discrepancy between the SNN's predictions and the ground-truth labels. This component is fundamental, as it directly optimizes the Student SNN for

the primary classification objective and ensures that the fine-tuning process remains grounded in solving the target task. For a batch of $N$ samples, the loss is described by

$$\mathcal{L}_{\text{task}} = -\frac{1}{N} \sum_{i=1}^{N} y_{i,\text{true}} \cdot \log(p_i), \quad (2)$$

where $y_{i,\text{true}}$ is the one-hot encoded ground-truth label for the $i$-th sample, and $p_i = \text{Softmax}(Z_{\text{SNN},i})$ is the predicted probability vector.

A critical detail in the SNN context is the derivation of the output logits, $Z_{\text{SNN},i}$. Unlike in an ANN where logits are static outputs, the SNN's final layer neurons produce discrete spike trains over $T$ time steps. To bridge this gap, we follow the standard practice of defining the logits as the time-averaged membrane potential (or equivalently, the total accumulated spike count) of these output neurons. This averaging process effectively integrates the temporal information encoded by the SNN into a static logit vector, $Z_{\text{SNN},i}$, which can then be processed by the Softmax function.

**Global Output Feedback ($\mathcal{L}_{\text{global}}$)** aligns the student's output distribution with that of the teacher, acting as the primary high-level corrective signal in our framework. We employ the standard KD loss (Hinton et al., 2015), which exploits the dark knowledge encoded in the relative probabilities of non-target classes. This information, which is absent in the hard one-hot labels used by $\mathcal{L}_{\text{task}}$, provides a much richer and more structured training signal. The central idea is to train the Student SNN to mimic the Teacher ANN's full output distribution, not just its final prediction. The temperature hyperparameter, $\tau > 1$, is critical to this process. By smoothing the softmax distributions of both networks, it raises the entropy of the probability vectors, thereby amplifying the small logit values corresponding to incorrect classes. This forces the SNN to learn why the teacher assigns certain relative probabilities to incorrect answers, capturing the teacher's understanding of inter-class similarity and ambiguity. In essence, while $\mathcal{L}_{\text{task}}$ trains the SNN to find the correct answer, $\mathcal{L}_{\text{global}}$ trains it to be correct in the same way as the superior ANN teacher. The loss is formulated as the Kullback-Leibler divergence between the softened outputs of the Teacher ANN and the Student SNN, which is described by

$$\mathcal{L}_{\text{global}} = D_{\text{KL}}(\sigma(Z_{\text{ANN}}/\tau) \parallel \sigma(Z_{\text{SNN}}/\tau)). \quad (3)$$

This loss acts as a powerful regularizer, transferring the teacher's generalization capabilities and preventing the SNN from merely overfitting to the hard labels provided by $\mathcal{L}_{\text{task}}$.

**Adaptive Local Constraint ($\mathcal{L}_{\text{local}}$)** explicitly aligns internal representations by introducing an adaptive weighting strategy. As illustrated in Fig. 3, a mini-batch is processed

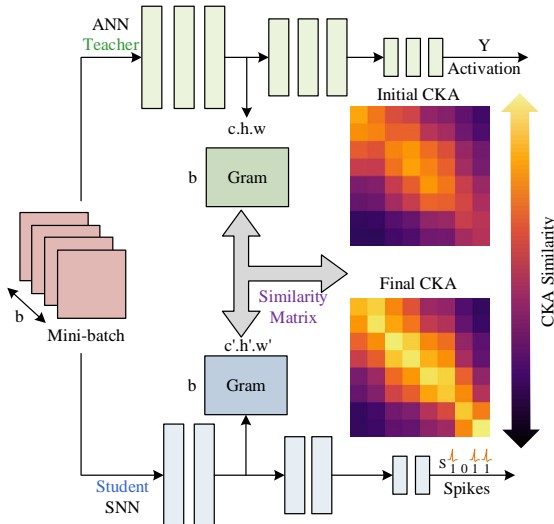

*Figure 3.* Visualization of CKA similarity matrices between the Teacher ANN and Student SNN.

concurrently by the Teacher and Student to compute the local CKA constraint. The central innovation is to move beyond uniform loss, prioritizing optimization on layers that exhibit the most severe representational damage from the initial open-loop conversion.

The local loss maximizes Centered Kernel Alignment similarity across selected layers $\mathcal{L}$:

$$\mathcal{L}_{\text{local}} = \sum_{l \in \mathcal{L}} w_l \cdot \left(1 - \text{CKA}\left(X_{\text{SNN}}^{(l)}, X_{\text{ANN}}^{(l)}\right)\right), \quad (4)$$

where $X_{\text{SNN}}^{(l)}$ and $X_{\text{ANN}}^{(l)}$ are feature maps at layer $l$.

The adaptive weighting factor $w_l$ is pre-calculated based on conversion fidelity:

$$w_l = \frac{1 - \text{CKA}_{\text{initial}}^{(l)}}{\sum_{j \in L}(1 - \text{CKA}_{\text{initial}}^{(j)})}. \quad (5)$$

Here, $(1 - \text{CKA}_{\text{initial}}^{(l)})$ quantifies the layer-wise information loss. By normalizing these scores, $\mathcal{L}_{\text{local}}$ dynamically reallocates optimization resources to the most misaligned representations.

We select CKA for this framework as it is superior to element-wise metrics (e.g., L2) for comparing heterogeneous networks. Its invariance to orthogonal transformations and isotropic scaling allows it to capture the underlying geometric similarity between ANN and SNN activation dynamics. Furthermore, CKA implicitly maps features into a reproducing kernel Hilbert space, relating to minimizing the Maximum Mean Discrepancy (Zhou et al., 2024).

*Table 3.* Comparison with State-of-the-Art ANN-to-SNN Conversion Methods on CIFAR-10 dataset.

| Architecture | Method | ANN Acc. | Altered | $T=2$ | $T=4$ | $T=8$ | $T=16$ | $T=32$ | $T=64$ |
|---|---|---|---|---|---|---|---|---|---|
| ResNet-18 | SNNC-AP (Li et al., 2021a) | 95.46 | ✓ | – | – | – | – | 94.78 | 95.30 |
| | OPI (Deng & Gu, 2021) | 96.51 | ✓ | 34.47 | 49.94 | 73.79 | 91.02 | 95.59 | 96.37 |
| | SlipReLU (Jiang et al., 2023) | 94.61 | ✓ | 93.11 | 93.97 | 94.59 | 94.92 | 95.18 | 95.07 |
| | QCFS (Bu et al., 2023) | 96.04 | ✓ | 75.44 | 90.43 | 94.82 | 95.92 | 96.08 | 96.06 |
| | ECL (Liu et al., 2025) | 96.39 | ✓ | 94.75 | 95.99 | 96.11 | 96.40 | 96.35 | 96.39 |
| | Twostage (Wang et al., 2023) | 96.41 | ✓ | 89.97 | 93.27 | 95.36 | 96.28 | 96.44 | 96.49 |
| | RMP (Han et al., 2020) | 96.60 | ✕ | 10.00 | 10.00 | 10.30 | 62.38 | 91.99 | 95.62 |
| | Spikingjelly (Fang et al., 2023) | 96.39 | ✕ | 10.00 | 10.00 | 49.05 | 86.97 | 93.90 | 95.54 |
| | **Ours** | 96.39 | ✕ | 90.05 | 93.58 | 95.33 | 96.26 | 96.38 | 96.41 |
| VGG-16 | SNNC-AP (Li et al., 2021a) | 95.72 | ✓ | – | – | – | – | 93.71 | 95.14 |
| | OPI (Deng & Gu, 2021) | 95.59 | ✓ | 68.37 | 88.03 | 93.05 | 94.74 | 95.42 | 95.51 |
| | SlipReLU (Jiang et al., 2023) | 93.02 | ✓ | 88.17 | 89.57 | 91.08 | 92.26 | 92.96 | 93.19 |
| | QCFS (Bu et al., 2023) | 95.52 | ✓ | 91.18 | 93.96 | 94.95 | 95.40 | 95.54 | 95.55 |
| | ECL (Liu et al., 2025) | 95.23 | ✓ | 93.53 | 95.15 | 95.41 | 95.39 | 95.48 | 95.43 |
| | Twostage (Wang et al., 2023) | 95.73 | ✓ | 91.71 | 94.06 | 95.26 | 95.55 | 95.71 | 95.78 |
| | RMP (Han et al., 2020) | 95.81 | ✕ | 10.00 | 10.00 | 10.00 | 26.89 | 81.13 | 93.34 |
| | Spikingjelly (Fang et al., 2023) | 94.98 | ✕ | 10.00 | 10.00 | 21.28 | 80.80 | 89.77 | 92.18 |
| | **Ours** | 94.98 | ✕ | 90.11 | 92.18 | 93.16 | 93.47 | 94.86 | 94.98 |

This alignment effect is visualized in Fig. 3: while the Initial CKA (post-conversion) shows weak correlation, the Final CKA exhibits a significantly brighter and sharper diagonal, confirming that our method successfully restores layer-by-layer feature fidelity.

### 3.4. Dual Alignment Mechanism

The composite loss in Eq. (1) orchestrates a multi-objective optimization, where $\alpha$ and $\beta$ regulate the interplay between task supervision, global output alignment ($\mathcal{L}_{\text{global}}$), and local feature constraints ($\mathcal{L}_{\text{local}}$). Theoretically, the efficacy of these terms is regime-dependent: at low latency (small $T$), $\mathcal{L}_{\text{global}}$ suffers from high-variance gradient estimates due to quantization noise (Deng et al., 2022), whereas $\mathcal{L}_{\text{local}}$ remains stable due to CKA's isotropic scaling invariance (Saha et al., 2022). Conversely, as $T$ increases, firing rates converge to accurate approximations, rendering $\mathcal{L}_{\text{global}}$ an unbiased estimator.

Guided by these dynamics, we calibrate hyperparameters to ensure balanced gradient contributions. We set $\alpha = 0.5$ to enforce equal weighting between ground-truth supervision and teacher knowledge, preventing overfitting to potential teacher errors. For $\beta$, we empirically calibrate it to normalize the gradient magnitudes of $\mathcal{L}_{\text{local}}$ relative to $\mathcal{L}_{\text{global}}$. This ensures that neither term dominates the optimization landscape, allowing the model to leverage local geometric stability and global semantic alignment synergistically across varying latency regimes. The specific impact of decoupling these terms is rigorously analyzed in our ablation study (Section 4.3).

## 4. Experimental Results

### 4.1. Implementation Details

Experiments were implemented using PyTorch and the SpikingJelly library (Fang et al., 2023). We employed standard data augmentation pipelines recommended by torchvision (maintainers & contributors, 2016) for both datasets to ensure robust generalization. For CIFAR-10, we used standard ResNet-18 (He et al., 2016) and VGG-16 (Simonyan & Zisserman, 2014) as fixed-weight Teacher ANNs. To evaluate scalability, we also extended our framework to the ImageNet dataset using ResNet-34. Initial ANN-to-SNN conversion employed SpikingJelly's ann2snn module with 99.9th percentile activation normalization. Student SNNs were fine-tuned for 50 epochs using the Adam optimizer and the composite loss function $\mathcal{L}_{\text{total}}$, which incorporates task loss, global distillation, and the adaptive CKA-based local loss. For the global distillation component ($\mathcal{L}_{\text{global}}$), the temperature hyperparameter $\tau$ was set to 2.0. All computations were performed on a single NVIDIA A100 SXM4 40GB GPU, without using distributed training. To clarify the computational overhead of the proposed closed-loop refinement, we additionally record the wall-clock time and peak GPU memory in the controlled ResNet-18/CIFAR-10 setting. The 50-epoch refinement takes about 1003 seconds at $T = 2$, with a peak GPU memory of 2.36 GB. In comparison, the conversion-only SpikingJelly ann2snn pipeline takes about 245 seconds and 0.32 GB peak GPU memory. Therefore, the proposed method introduces additional post-conversion cost, but this cost is a one-time refinement cost starting from an already converted SNN, rather than training an SNN from scratch.

*Table 4.* Performance comparison with methods without altering the source ANN architecture on CIFAR-10 and IMAGENET.

| Method | Training | Archi. | $T$ | Acc. |
|---|---|---|---|---|
| **CIFAR-10** | | | | |
| (Rathi & Roy, 2020) | STBP | VGG16 | 5 | 92.70 |
| (Rathi et al., 2020) | Hybrid | VGG16 | 200 | 92.02 |
| **Ours** | Hybrid | VGG16 | 8 | **93.16** |
| (Rathi & Roy, 2020) | STBP | ResNet20 | 5 | 91.78 |
| (Li et al., 2021b) | STBP | ResNet18 | 4 | 93.66 |
| (Rathi et al., 2020) | Hybrid | ResNet20 | 250 | 92.22 |
| **Ours** | Hybrid | ResNet18 | 8 | **95.33** |
| **IMAGENET** | | | | |
| (Han et al., 2020) | ANN2SNN | ResNet34 | 256 | 55.65 |
| (Fang et al., 2023) | ANN2SNN | ResNet34 | 64 | 39.00 |
| (Rathi et al., 2020) | Hybrid | ResNet34 | 256 | 61.48 |
| **Ours** | Hybrid | ResNet34 | 16 | **55.13** |

*Table 5.* Ablation study on CIFAR-10, showing accuracy (%) at different time steps.

| Configuration | Accuracy (%) at Time Steps $T$ | | | | | |
|---|---|---|---|---|---|---|
| | $T=2$ | $T=4$ | $T=8$ | $T=16$ | $T=32$ | $T=64$ |
| Baseline ($\alpha = 0$) | 85.08 | 93.04 | 95.10 | 95.79 | 96.13 | 96.14 |
| + $\mathcal{L}_{\text{global}}$ ($\beta = 1.0$) | 87.68 | 93.32 | 94.92 | 96.10 | **96.38** | **96.41** |
| + $\mathcal{L}_{\text{local}}$ ($\beta = 0.0$) | 88.86 | 93.50 | 95.27 | 95.86 | 96.13 | 96.18 |
| **Ours** ($\beta = 0.5$) | 85.09 | 93.15 | **95.33** | **96.26** | 96.23 | 96.32 |
| **Ours** ($\beta = 0.2$) | **90.05** | **93.58** | **95.33** | 96.03 | 96.29 | 96.27 |

Source: ResNet-18 (ANN: 96.39%) with ImageNet-V1 weights. $\alpha = 0.5$ for non-baseline rows.

## 4.2. Comparison with SoTA Methods

As detailed in Section 4.1, we utilize standard architectures to ensure reproducibility. Since the source ANN establishes the performance upper bound in ANN-to-SNN conversion, we prioritize assessing conversion fidelity, defined as the accuracy gap relative to these specific baselines, rather than focusing exclusively on absolute accuracy comparisons.

Table 3 compares our method with state-of-the-art ANN-to-SNN conversion approaches on CIFAR-10. In this comparison, our method demonstrates a clear advantage. At the low time step $T = 8$, the SpikingJelly baseline yields near-random accuracy, while our conversion remains highly accurate. As the time window increases, our method achieves virtually lossless conversion at $T = 32$. Specifically, the converted ResNet-18 reaches 96.38% accuracy versus 96.39% for the source ANN. These results indicate that our dual-alignment strategy achieves fidelity comparable to approaches relying on structural modifications (e.g., QCFS) without altering the original architecture.

Table 4 benchmarks our model against state-of-the-art techniques that preserve the source ANN architecture. HCSB (Rathi et al., 2020), a representative hybrid strategy, typically requires large time windows to stabilize. In contrast, our VGG-16 attains 93.16% at $T = 8$, outperforming HCSB despite the latter's longer inference window. Similarly, our ResNet-18 achieves 95.33% at $T = 8$, surpassing pure STBP methods trained from scratch, such as Dspike (Li et al., 2021b) (93.66% at $T = 4$). We also extend the evaluation to ImageNet. Restricted to $T = 16$ due to resource constraints, our ResNet-34 achieves 55.13% accuracy. This performance suggests that the proposed framework operates effectively in low-latency regimes where traditional hybrid methods typically struggle.

## 4.3. Ablation Study

We conduct a comprehensive ablation study to quantify the individual and synergistic contributions of the global ($\mathcal{L}_{\text{global}}$) and local ($\mathcal{L}_{\text{local}}$) loss components. Our analysis systematically deconstructs the composite loss function Eq. (1) by evaluating the configurations defined in Sec. 3.4. We establish a Baseline trained solely with the standard task loss $\mathcal{L}_{\text{task}}$ ($\alpha = 0$). Subsequently, as noted in Table 5, we evaluate configurations with $\alpha = 0.5$, comparing a global-only setting ($\beta = 1.0$), a local-only setting ($\beta = 0.0$), and our full models with intermediate $\beta$ values. The effectiveness of each configuration is measured by the SNN's top-1 accuracy on CIFAR-10 across various inference time steps $T$, assessing both final performance and convergence efficiency.

The ablation in Table 5 reveals a clear, time-dependent trade-off between the two feedback mechanisms. At small time steps, such as $T = 2$ and $T = 4$, the SNN's sparse spike train is a high-variance approximation of the teacher's continuous logit distribution. This renders $\mathcal{L}_{\text{global}}$ an unstable, and potentially misleading, gradient source. The $\mathcal{L}_{\text{local}}$ loss, however, is less sensitive to this temporal quantization as it provides a structural constraint on representational geometry. This explains why the local-only model ($\beta = 0.0$) outperforms the global-only model ($\beta = 1.0$) at $T = 4$, and why our full model with high local weight (Ours, $\beta = 0.2$) yields the best low-$T$ accuracy. As $T$ increases, the aggregated spike counts become a faithful, low-variance approximation of the teacher's logits. Consequently, $\mathcal{L}_{\text{global}}$ becomes a high-quality optimization target. This is evident at $T = 64$, where performance monotonically improves with the global loss weight: Accuracy increases from 96.18% in the local-only setting ($\beta = 0.0$) to the maximum of 96.41% in the global-only setting ($\beta = 1.0$). The baseline ($\alpha = 0$) consistently underperforms, while our full model's strong performance across all $T$ validates this synergistic design, achieving high accuracy in both low-latency ($T \leq 8$) and high-fidelity ($T = 64$) regimes.

Fig. 5 visualizes the impact of our framework on internal feature representations across network depth. We omit $T = 32$

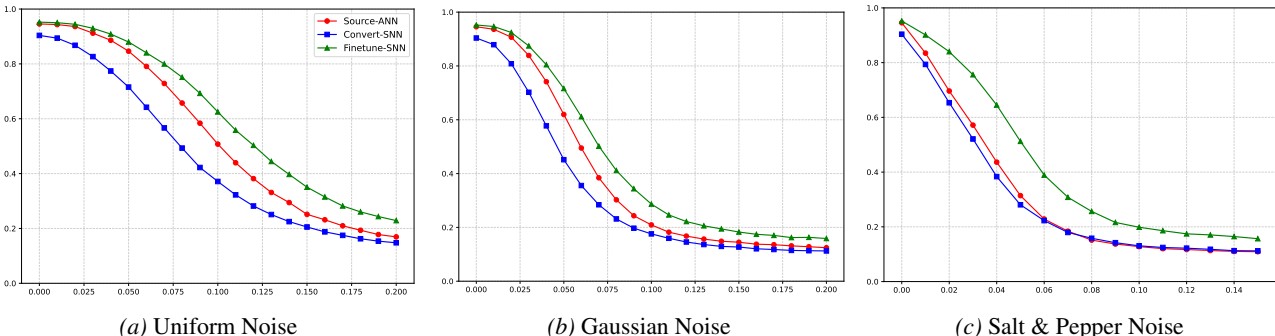

*(a)* Uniform Noise          *(b)* Gaussian Noise          *(c)* Salt & Pepper Noise

*Figure 4.* Noise robustness evaluation. Accuracy curves of the Source ANN, baseline converted SNN (Convert-SNN), and our fine-tuned SNN (Finetune-SNN) under increasing intensities of Uniform, Gaussian, and Salt & Pepper noise.

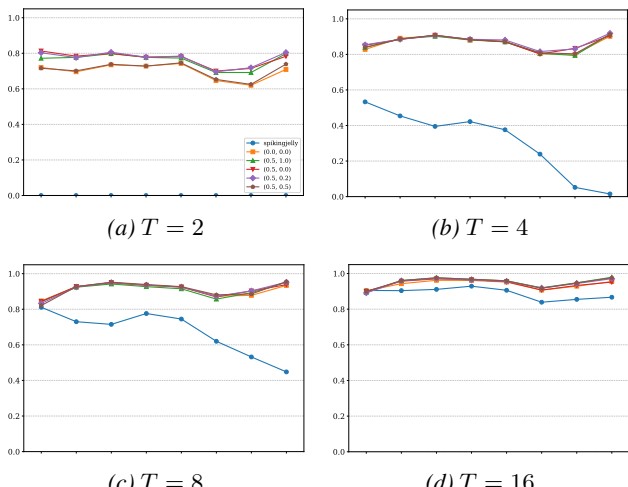

*(a)* $T = 2$          *(b)* $T = 4$

*(c)* $T = 8$          *(d)* $T = 16$

*Figure 5.* Layer-wise CKA similarity between the fine-tuned SNN and the source ANN across different time steps ($T$).

and $T = 64$ from this visualization, as their initial conversion already yields high CKA scores, making the subsequent changes less visually informative. The SpikingJelly baseline conversion (blue line) demonstrates a severe degradation of the teacher ANN's feature representation during open-loop conversion, with CKA values dropping to zero at $T = 2$. Moreover, the baseline model exhibits a significant decline in CKA similarity with increasing network depth across all time steps, highlighting the impact of cascading error propagation. In contrast, our fine-tuned models (all lines except the baseline) significantly mitigate this degradation. Notably, our local loss, $\mathcal{L}_{\text{local}}$, not only preserves feature similarity but also promotes its recovery. For instance, at $T = 8$ and $T = 16$, the CKA similarity of the final residual blocks often slightly exceeds that of the initial blocks. This suggests that the local constraint actively corrects for feature distortions accumulated in deeper layers. While the CKA values show minimal divergence at $T \geq 32$, they are still further improved by our closed-loop fine-tuning.

## 4.4. Robustness to Noisy Inputs

To validate the resilience of our framework, we evaluated the SNNs against the source ANN under three common noise types at increasing intensities: Uniform, Gaussian, and Salt & Pepper, as shown in Fig. 4.

Our Finetune-SNN consistently and significantly outperforms the baseline Convert-SNN across all noise types. For instance, under Gaussian noise at an intensity of 0.1 (Fig. 4b), the baseline SNN's accuracy collapses to ≈20%, whereas our model retains over ≈40% accuracy. Furthermore, our framework significantly closes the performance gap to the Source-ANN, indicating that our closed-loop process improves generalization beyond correcting static conversion errors. This effect is most pronounced in the Salt & Pepper noise test (Fig. 4c), where our Finetune-SNN not only matches but slightly surpasses the Source-ANN's accuracy at low-to-moderate noise levels (0.02 to 0.05 intensity). This strongly suggests our combined feedback acts as an effective regularizer, yielding a final model with superior robustness to input perturbations.

## 5. Conclusion and Discussion

In this work, we proposed a closed-loop fine-tuning framework to address the accuracy degradation and temporal inefficiency in ANN-to-SNN conversion, which we attribute to accumulating conversion errors across network depth. Our approach integrates spatio-temporal backpropagation within a teacher-student paradigm, employing a dual-alignment strategy that combines global output feedback with adaptive CKA-based local constraints. This design effectively mitigates layer-wise feature drift without altering the source architecture. Empirical results demonstrate that our framework achieves lossless accuracy with significantly reduced inference latency. Notably, the fine-tuned SNNs exhibit superior robustness to input noise, confirming the strong regularization effect of the proposed method. It is worth noting that this work mainly focuses on improving con-

version fidelity and reducing the required inference time steps. Although SNNs have potential energy advantages due to event-driven spike-based computation, we do not claim hardware-measured end-to-end energy savings in this study. A detailed energy evaluation on neuromorphic hardware will be considered in future work. Future work will extend this paradigm to event-based vision tasks and investigate its scalability to more complex architectures, such as Spiking Transformers.

## Acknowledgements

This work was supported in part by the National Natural Science Foundation of China under Grant Nos. 62441234 and 62532005.

## Impact Statement

This paper aims to advance energy-efficient machine learning by improving high-fidelity and low-latency ANN-to-SNN conversion. The experiments are conducted on standard image classification benchmarks and do not involve human subjects, private data, or personally identifiable information. The potential risks are mainly inherited from downstream applications of converted models, including dataset bias, distribution shift, unreliable predictions, or misuse in visual recognition systems. Although SNNs may offer potential energy advantages, this paper does not claim hardware-measured end-to-end energy savings. We do not identify additional ethical concerns beyond these general considerations.

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
