# OpenReview forum: "High-Fidelity ANN-to-SNN Conversion via Closed-Loop CKA Distillation"
_ICML.cc/2026/Conference — ICML 2026 regular_

### Official Review · Reviewer_pcsp · 2026-03-11

**Soundness:** 3
**Presentation:** 3
**Significance:** 3
**Originality:** 3
**Overall Recommendation:** 4
**Confidence:** 3

**Summary:**

This paper studies ANN-to-SNN conversion under the fidelity-latency trade-off caused by open-loop error accumulation in deep spiking networks. To address this issue without modifying the source ANN, the authors propose a closed-loop fine-tuning framework that first converts a pretrained ANN into an SNN and then fine-tunes the converted SNN using a teacher-student setup with two alignment signals: global output distillation via KL divergence and adaptive local feature alignment via CKA. The key idea is that local alignment stabilizes optimization in low-latency regimes, while global alignment becomes more effective at larger time steps, and the method combines both to reduce conversion error across layers. Experiments on CIFAR-10 and ImageNet show that the approach substantially improves low-latency conversion fidelity, achieves near-lossless accuracy for ResNet-18 on CIFAR-10 at moderate time steps, and also improves robustness to noisy inputs compared with standard converted SNN baselines.

**Compliance With Llm Reviewing Policy:**

Affirmed.

**Final Justification:**

After considering both the paper and the authors’ rebuttal, I am raising my score from Weak Reject to Borderline Accept. The paper has clear strengths: it targets an important ANN-to-SNN conversion limitation, presents a technically coherent closed-loop refinement framework, and shows strong low-latency CIFAR-10 results with a meaningful time-dependent pattern between local and global alignment. The rebuttal also addressed two of my main concerns in a substantive way, namely the scalability/headroom question and the role of adaptive weighting beyond CKA itself. I still have some reservations about originality, since the contribution still reads more as a careful organization of known ingredients than a fundamentally new principle, and I think the sensitivity of the loss balance across latency regimes remains a real caveat. Even so, the rebuttal improved my overall assessment enough that I now see the paper as a borderline case leaning positive rather than a clear weak reject.

**Key Questions For Authors:**

- The method combines several ingredients that already appear in the post-conversion fine-tuning literature, including STBP, KD-based output matching, and intermediate feature alignment. Could the authors clarify more precisely what they view as the main technical novelty beyond combining these components, especially relative to prior KD- and CKA-based post-conversion approaches?
- The ablation suggests that the best loss balance depends quite strongly on the latency regime: local alignment is stronger at low T, while global distillation is stronger at high T, and the proposed combined setting is not consistently the best-performing variant across all time steps. How sensitive is the method to the choice of β, and is there a principled way to adapt β as a function of T rather than fixing it manually?
- The CIFAR-10 results are strong, but the large-scale evidence is more limited. On ImageNet, the evaluation appears restricted to ResNet-34 at T=16 due to resource constraints. Could the authors provide more discussion on how the method scales with larger T, stronger architectures, or longer fine-tuning, and whether the current ImageNet result is representative of the method’s full potential?
- The adaptive local weighting is based on the initial layer-wise CKA loss after conversion. Did the authors try updating these weights during training instead of fixing them from the initial conversion state? It would be helpful to know whether the gains come mainly from using CKA itself, from the adaptive weighting, or from the specific initialization of those weights.

**Limitations:**

yes

**Strengths And Weaknesses:**

# Strengths
- Closed-Loop CKA Distillation targets an important limitation of ANN-to-SNN conversion, namely the fidelity-latency trade-off caused by open-loop error accumulation, and the overall idea is well motivated and technically coherent.
- Closed-Loop CKA Distillation shows strong empirical results on CIFAR-10, especially in the low-latency regime. For ResNet-18, it achieves near-lossless conversion at T=32 (96.38% vs. 96.39% for the source ANN), and its low-T results at T=8 are much stronger than vanilla conversion baselines that do not alter the source architecture.
- The dual-alignment design is probably the most convincing part experimentally. The ablation suggests a meaningful time-dependent pattern: local alignment helps more at very small T, while global distillation becomes more useful at larger T. This gives some support to the central intuition, rather than making the method look like an arbitrary combination of losses.

# Weaknesses
- Closed-Loop CKA Distillation appears somewhat limited in novelty. The method is positioned within post-conversion fine-tuning with STBP, KD, and intermediate feature alignment, so the contribution reads more like a careful combination and weighting of known ingredients than a fundamentally new conversion principle.
- The empirical case for Closed-Loop CKA Distillation is strongest on CIFAR-10, but the evidence on larger-scale settings is less convincing. On ImageNet, the reported ResNet-34 result is 55.13% at T=16, which is below the listed hybrid baseline of 61.48% at T=256. Although the latency is much lower, the large-scale advantage is still less clear than the CIFAR-10 story suggests.
- The ablation results also show that the full version is not consistently best across regimes. At high T, the global-only variant performs best, while at low T the strongest result comes from a more locally weighted variant (β=0.2), not the symmetric combined version. This suggests some sensitivity to the loss balance, and the benefit of the proposed combination does not look fully robust.

---

> ### Author Rebuttal · Authors · 2026-03-31
>
> Thanks for your careful review and for the thoughtful questions. We appreciate your recognition that the paper addresses an important ANN-to-SNN limitation, namely the fidelity-latency trade-off caused by open-loop error accumulation. We also appreciate your positive assessment of the strong low-latency CIFAR-10 results, especially your observation that the dual-alignment design shows a meaningful time-dependent pattern rather than an arbitrary combination of losses. Unless otherwise stated, the additional rebuttal experiments below are on ResNet-18/CIFAR-10.
>
> Q1：The main novelty, in our view, is not KD or CKA by themselves, but how they are organized to address a specifically SNN-side problem: representation drift introduced by open-loop conversion under low-latency constraints. Our method treats post-conversion refinement as a closed-loop correction process in which global KD preserves output semantics, while local CKA repairs layer-wise distortion according to the initial conversion damage profile. The ablations further show that these two signals play different roles across latency regimes. We therefore view the contribution as a representation-aware refinement framework for low-latency ANN-to-SNN conversion, rather than a generic combination of existing losses.
>
> Q2：We agree that the best loss balance depends on the latency regime. We view this as a main empirical finding rather than a weakness. Table 5 already shows a clear pattern: local alignment is more beneficial at low T, while global distillation becomes stronger as T increases. We therefore do not claim that one fixed β is optimal for all regimes. Instead, our results suggest a simple interpretation: lower β is more suitable when conversion noise is severe, while larger β becomes more appropriate as the spike-based approximation improves. This is why we use β=0.2 in the low-latency setting.
>
> Q3：The reported 55.13% top-1 accuracy at T=16 should be viewed as a low-latency checkpoint rather than the upper bound of the framework. To test scalability beyond this setting, we extended the ResNet-34/ImageNet evaluation to T=64 and obtained 60.83% top-1 accuracy. This shows that the method retains substantial headroom beyond the submitted T=16 result and directly addresses the related ImageNet scalability concern raised by Reviewer Tf8V.
>
> At the same time, our primary objective is to improve conversion fidelity under strict low-latency constraints. Substantially increasing T or prolonging the fine-tuning process incurs massive GPU memory and computational overhead. Crucially, it diminishes the fundamental energy efficiency advantage of SNNs over ANNs; as detailed in our response to Reviewer MC9t, the estimated ANN/SNN forward energy ratio drops drastically from approximately 38.9x at T=2 to merely 2.8x at T=32. Furthermore, scaling to stronger architectures like Transformers introduces coding-level paradigm shifts, such as temporal-coding spiking neurons and spiking attention mechanisms. Incorporating these elements would inherently violate the core premise of our framework, which is to achieve seamless conversion without modifying the source ANN architecture.
>
> Q4. We investigated in two ways whether the gains come from CKA itself, adaptive weighting, or the initialization of the weights.
>
> First, we tested dynamic updating of the CKA-based layer weights during training instead of fixing them from the initial conversion state. For the full static-versus-dynamic comparison, including the memory cost across different T, please see our response to Reviewer Tf8V. The conclusion is consistent: dynamic updating gives only marginal accuracy gains but incurs a very large memory cost. For example, on ResNet-18/CIFAR-10, the dynamic variant improves over the static one only from 90.05% to 90.21% at T=2, and from 96.29% to 96.33% at T=32.
>
> Second, to isolate the role of adaptive weighting itself, we compared uniform local weighting against our adaptive $w_l$ under the same 50-epoch fine-tuning. At T=2, the uniform version reaches 87.57%, while the adaptive version reaches 90.05%. This shows that the gain does not come from CKA alone: adaptive weighting based on the initial conversion damage profile provides an additional benefit over a uniform local constraint.
>
> Taken together, these results suggest that local CKA is useful, adaptive weighting is more effective than uniform weighting, and fixing $w_l$ from the initial post-conversion state is a practical choice, since dynamic updating brings only very small gains at high cost.
>
> Thank you for your constructive feedback and time.

---

> > ### Author Rebuttal · Reviewer_pcsp · 2026-04-03
> >
> > Thank you for the thoughtful rebuttal. It addressed several of my main concerns and improved my evaluation of the paper. In particular, the added evidence on ImageNet at larger T helps clarify that the reported T=16 result was not the ceiling of the method, and the additional analyses on static vs. dynamic layer weighting and uniform vs. adaptive weighting make the role of the adaptive CKA design much clearer. I still view the main novelty as more of a well-motivated integration and refinement of existing post-conversion ingredients than a fundamentally new conversion principle, and I also think the method remains somewhat sensitive to the latency regime through the choice of β. That said, the rebuttal makes the empirical story more convincing, especially for low-latency conversion, and overall it strengthened my confidence in the paper.

---

> > > ### Author Response · Authors · 2026-04-03
> > >
> > > Thank you very much for your thoughtful reply and for your improved evaluation of our paper. We sincerely appreciate your careful reading, fair assessment, and encouraging feedback. We are glad that our rebuttal helped clarify the key points of the work.

---

### Official Review · Reviewer_Tf8V · 2026-03-12

**Soundness:** 3
**Presentation:** 3
**Significance:** 2
**Originality:** 2
**Overall Recommendation:** 4
**Confidence:** 4

**Summary:**

This manuscript presents a closed-loop fine-tuning framework for the conversion of ANN to SNN, aiming to address the performance degradation and high latency issues caused by open-loop error accumulation. The core innovation lies in a dual alignment mechanism that combines global KL divergence for output distillation and local CKA constraints for feature-level calibration. By adaptively adjusting the weights of these constraints based on the initial conversion loss, this method effectively stabilizes the representations in the low-latency mode, where global gradients are typically unstable. Experimental results demonstrate high fidelity, with the SNN achieving nearly the same accuracy as the source ANN on the CIFAR dataset while significantly reducing the time steps. Additionally, the framework enhances the robustness of the converted SNN to input noise through its regularization effect. Overall, this approach offers a scalable and efficient solution for deploying high-performance SNNs using off-the-shelf ANN models.

**Compliance With Llm Reviewing Policy:**

Affirmed.

**Final Justification:**

The authors have adequately addressed the concerns I raised. ANN-to-SNN conversion is a meaningful topic with practical relevance, and the proposed method demonstrates significant improvements on public datasets. Therefore, I have increased my rating.

**Key Questions For Authors:**

1. This paper claims to have provided an efficient conversion method that can avoid the high computational cost involved in training SNN from scratch. However, the "closed-loop" refinement stage actually requires the use of STBP technology to fine-tune the entire model for 50 epochs. This process, which requires a large amount of memory and is time-consuming, is actually equivalent to an expensive re-training stage, which is directly contrary to the main goal of quickly and costlessly converting artificial neural networks into spatial-temporal neural networks.
2. This framework introduces an "adaptive" weight to balance the local CKA constraints between different network layers. Its main drawback lies in that this weight is fixed and unchanged based solely on the initial transformation loss before the fine-tuning begins. Using a static one-time metric to guide a dynamic 50-epochs training process is mathematically inconsistent because it completely ignores the changes in the internal feature maps and error distribution of SNN over time.
3. Although this method can achieve remarkable accuracy on smaller datasets like CIFAR, it performs poorly when dealing with complex tasks. In the ImageNet evaluation, the transformed ResNet-34 model achieved only 55.13% accuracy at T=16. This significant performance gap indicates that the proposed dual alignment mechanism loses its effectiveness when applied to large-scale, high-dimensional data, thereby limiting its practical application value.

**Limitations:**

No.
Suggestions:
1. The authors can discuss the effects of other backbone network conversions.
2. Why not increase the appropriate number of steps to achieve the optimal result on the Imagenet dataset?

**Strengths And Weaknesses:**

Strengths
Soundness: The methodology is technically robust, addressing the "open-loop" error accumulation of traditional conversion by using a "closed-loop" fine-tuning method. The use of CKA is well-motivated as a metric for feature-map consistency, and the experiments on CIFAR datasets show empirical results that nearly close the gap between ANN and SNN performance.
Originality: This paper ingeniously combines two existing techniques - knowledge distillation and CKA. It introduces an adaptive weighting mechanism that balances local and global constraints based on the transformation loss. This innovative and effective insight is of good value.
Significance: This research has addressed the crucial "trade-off between fidelity and latency" issue in neuromorphic computing. This framework can achieve high-performance operation with low latency, and does not require re-training of the original artificial neural network. Therefore, it has high practical application value for deploying existing deep learning models onto energy-efficient pulse-based hardware.
Presentation: The core narrative is clear.
Weaknesses
Originality: Although the application of CKA to the SNN conversion method is relatively new, this study improves the traditional "teacher-student" training process by focusing on the way knowledge is transferred, rather than proposing a new method.
Significance: One drawback lies in the "hidden costs" during the closed-loop fine-tuning process. One of the major advantages of converting an artificial neural network into a neural network model is its "no-training" feature. By introducing a fine-tuning stage, this approach is more akin to "direct training" or "hybrid training", which might reduce its appeal to users who are seeking immediate and zero-cost deployment.
Presentation: The paper mentions an adaptive weighting mechanism, but the discussion on how it affects the final performance and the relationship between the initialization of these weights or the selection of specific layers is rather limited. This information is crucial for achieving reproducibility and applying this method to different architectures.

---

> ### Author Rebuttal · Authors · 2026-03-31
>
> Thanks for your careful review and for the time you spent evaluating our work. We appreciate your positive remarks about the robustness of the closed-loop methodology, the value of the dual alignment mechanism, and the strong empirical results in low-latency settings. Unless otherwise stated, all additional rebuttal experiments are conducted on ResNet-18/CIFAR-10. To ensure a cleanly controlled comparison, we adopt the setting from our ablations, corresponding to the "Ours ($\beta$ = 0.2)" row in Table 5 of the main paper.
>
> 1. On whether the closed-loop refinement stage is essentially an expensive retraining step
>
> We agree that our method introduces extra cost beyond direct conversion, but it is still much lighter than training an SNN from scratch. It starts from an already converted SNN and performs a short refinement stage specifically to recover the low-timestep fidelity lost in open-loop conversion.
>
> In our experiment, the 50-epoch refinement takes about 1003 s at T=2, with peak GPU memory of 2.36 GB. By comparison, the conversion-only SpikingJelly `ann2snn` pipeline takes about 245 s and only 0.32 GB peak GPU memory, since it mainly performs calibration for threshold / voltage normalization. However, the cheaper conversion-only baseline has very limited low-T fidelity: it achieves only 10.00% at T=2 and 93.90% at T=32, whereas our method reaches 90.05% at T=2 and 96.38% at T=32.
>
> So the trade-off is clear: pure conversion is cheaper, but it fails exactly in the low-latency regime that motivates this work. Our method adds moderate post-conversion cost, and that cost is used to close most of the fidelity gap left by direct conversion.
>
> 2. On why the CKA-based layer weights are fixed rather than updated dynamically
>
> In our method, $w_l$ in Eq. (5) is computed once from the initial post-conversion mismatch and then kept fixed during refinement. The purpose is to identify, at the start of training, which layers are most distorted by conversion, and to use this as a stable layer prior.
>
> We also implemented a dynamic variant that recomputes the CKA-based weights during training. The result is summarized below:
>
> | TimeStep | Static Accuracy | Dynamic Accuracy | Static Memory (GB) | Dynamic Memory (GB) | Firing Rate | Total Time for 50 Epochs (min) |
> |---|---:|---:|---:|---:|---:|---:|
> | T=2  | 90.05% | 90.21% | 2.36 | 12.68 | 6.58% | 16.73 |
> | T=4  | 93.58% | 93.70% | 3.15 | 16.19 | 8.06% | 28.13 |
> | T=8  | 95.33% | 95.47% | 4.86 | 23.81 | 7.38% | 52.88 |
> | T=16 | 96.03% | 96.11% | 6.09 | 28.68 | 6.56% | 110.97 |
> | T=32 | 96.29% | 96.33% | 7.69 | 35.87 | 6.75% | 290.43 |
>
> As demonstrated above, dynamic CKA-based weights yields only marginal accuracy improvements (e.g., +0.16% at T=2 and +0.04% at T=32), yet incurs a massive GPU memory overhead across all T. This suggests that most of the useful layer-priority signal is already captured at initialization, while repeated recomputation is expensive and provides little additional benefit. For this reason, we adopt the static form in Eq. (5), which offers a strictly superior trade-off between efficiency and fidelity.
>
> 3. On ImageNet scalability and whether T=16 reflects the method’s full potential
>
> We agree that the original ImageNet evidence in the submission was limited. The reported 55.13% top-1 accuracy at T=16 should be viewed as a low-latency checkpoint rather than the full potential of the framework. To further examine scalability, we additionally evaluated ResNet-34 on ImageNet at T=64 and obtained 60.83% top-1 accuracy. This result suggests that the method retains further headroom beyond the submitted T=16 setting.
>
> At the same time, the main objective of this work is not simply to increase T until accuracy becomes optimal, but to improve fidelity under practically relevant low-latency settings, where ANN-to-SNN conversion is most challenging. Increasing T introduces a substantial increase in memory and computational cost because T acts as an additional temporal dimension in SNN simulation. This trend is also consistent with our rebuttal experiments on ResNet-18/CIFAR-10, where larger T leads to much longer refinement time and significantly higher memory usage.
>
> Regarding the suggestion about other backbone conversions, we agree this is an important direction. In the current paper, we focus on CNN backbones because CNN-to-SNN conversion remains the most standard and controlled setting for studying the fidelity–latency trade-off caused by open-loop conversion errors. Extending the same closed-loop refinement idea to other backbone families, especially Transformer-based models, is valuable but also introduces additional operator-level challenges beyond standard CNN conversion, such as attention-related nonlinear operators. We therefore view this as an important extension rather than a missing component of the present study.
>
> We thank you again for the helpful comments.

---

> > ### Author Rebuttal · Reviewer_Tf8V · 2026-04-01
> >
> > The authors have addressed most  of the concerns I raised. But I still have a question: At what level of conversion accuracy can this be considered acceptable （more than 60% or 80%）?

---

> > > ### Author Response · Authors · 2026-04-03
> > >
> > > Thank you very much for your supportive attitude toward our paper.
> > >
> > > **Q1: At what level of conversion accuracy can this be considered acceptable (e.g., above 60% or 80%)?**
> > >
> > > In our view, there is no universal fixed threshold, such as 60% or 80%, for ANN-to-SNN conversion. A more appropriate criterion is the accuracy–timestep trade-off. Under a relatively low timestep budget, while preserving the energy-efficiency advantage of SNNs over ANNs, the converted SNN should retain as much of the original ANN accuracy as possible. The evaluation can be understood in terms of the accuracy gap, i.e., \(Acc(ANN) - Acc(SNN)). At present, we believe that on CIFAR tasks, the accuracy gap should be close to 5%, whereas on ImageNet, an accuracy loss within 20% can still be considered acceptable. Moreover, this standard is not static, but will continue to rise as the SNN field advances.
> > >
> > > If you feel our paper is worthy of support, we would greatly appreciate a brief final justification. Thank you again for your time and support.

---

### Official Review · Reviewer_MC9t · 2026-03-13

**Soundness:** 3
**Presentation:** 3
**Significance:** 3
**Originality:** 2
**Overall Recommendation:** 4
**Confidence:** 5

**Summary:**

The paper addresses the accuracy loss and latency trade-off in ANN-to-SNN conversion, where converting ANNs into SNNs often requires many timesteps to recover the original ANN performance due to accumulated spike quantization errors.
To mitigate this, the authors propose a post-conversion fine-tuning framework that keeps the original ANN unchanged. After converting the ANN to an SNN, the model is optimized using a teacher-student distillation setup, where the ANN acts as a frozen teacher guiding the SNN.
The method introduces a dual alignment mechanism: global alignment, which matches the output distributions of the ANN and SNN using knowledge distillation, and a local alignment, which encourages similarity between intermediate feature representations using CKA, with adaptive weights emphasizing layers most affected by conversion.
Experiments on CIFAR-10 show that the proposed method significantly improves conversion fidelity, achieving near-ANN accuracy with fewer timesteps and better performance than several existing conversion methods that do not modify the source model.

**Compliance With Llm Reviewing Policy:**

Affirmed.

**Final Justification:**

I appreciate the author’s rebuttal response; however, I will keep my original score because my main concerns remain insufficiently addressed.

**Key Questions For Authors:**

None

**Limitations:**

No.
The authors could acknowledge several practical limitations, such as the additional computational cost introduced by post-conversion fine-tuning, the limited evaluation on small-scale datasets, and the lack of direct measurements of energy efficiency or deployment on neuromorphic hardware. It would also be useful to discuss potential scalability challenges when applying the method to larger architectures or tasks.

**Strengths And Weaknesses:**

Strengths:
The paper addresses a well-motivated problem in ANN-to-SNN conversion.
The proposed method is practical because it improves the converted SNN without modifying the original ANN architecture or training process, allowing it to work with standard pre-trained models.
The training objective is well structured, combining output-level knowledge distillation with intermediate feature alignment to guide the SNN during fine-tuning. In particular, the adaptive weighting of feature alignment using layer-wise CKA similarity is a reasonable design choice.
Experimentally, the method demonstrates clear improvements over baseline conversion approaches, especially in low-timestep regimes.

Weaknesses:
The paper's novelty is somewhat limited, as the proposed method mainly combines existing techniques such as distillation, feature alignment, and post-conversion fine-tuning rather than introducing a fundamentally new learning mechanism.
The experimental evaluation is also relatively limited, with most results reported on CIFAR-10 and only a brief experiment on ImageNet, which makes it difficult to assess scalability to larger and more complex tasks.
Although the paper emphasizes latency improvements, the evaluation primarily reports accuracy across timesteps and doesn't measure practical efficiency metrics such as runtime, spike activity, or energy consumption.
The proposed approach requires additional fine-tuning of the converted SNN using spatio-temporal backpropagation, which introduces extra training cost compared to standard conversion pipelines.
Some design choices, including the selection of several hyperparameters, are also largely empirical and not thoroughly analyzed.

---

> ### Author Rebuttal · Authors · 2026-03-30
>
> We sincerely thank you for the positive assessment of our work. We also appreciate your recognition of the practicality of keeping the source ANN unchanged and the strong gains in the low-timestep regime. We agree that the contribution of this work is not a fundamentally new learning mechanism. Instead, we see the main contribution as a practical closed-loop post-conversion refinement framework for ANN-to-SNN conversion. It combines global KD for output-level guidance and adaptive local CKA for correcting layer-wise representation drift caused by open-loop conversion. Our ablations further suggest that these two signals play different roles across latency regimes: local alignment is more important at very small T, while global distillation becomes more effective as T grows.
>
> For clarity, unless otherwise stated, all additional rebuttal experiments are conducted on ResNet-18/CIFAR-10 for fast and controlled verification. In this setting, the timestep T acts as an additional temporal dimension, and increasing T substantially increases memory usage. With static CKA layer weights $w_l$ in Eq. (5), peak memory grows from 2.36 GB at T=2 to 7.69 GB at T=32. This is also why larger-T experiments, especially on ImageNet, are much harder in practice on a single A100 40GB GPU.
>
> Regarding scalability, we also evaluated ResNet-34 on ImageNet at T=64 and obtained 60.83% top-1 accuracy. Because SNN memory grows strongly with T, this result used batch size 4 and should therefore be viewed as a strong accuracy-oriented scalability check, demonstrating that the proposed dual-alignment mechanism scales effectively to complex datasets and larger architectures.
>
> We also agree that practical efficiency should be better documented. Since we do not have access to neuromorphic hardware for real deployment-time latency/energy measurement, we only provide a simple compute-side energy proxy as a reference, following prior SNN work (Rathi & Roy, 2020), which assumes that each ANN MAC consumes 4.6 pJ and each SNN AC consumes 0.9 pJ. Under this commonly used assumption, our measured operation counts indicate that the SNN has a much more favorable compute-energy profile at small T, and the advantage becomes smaller as T increases: the estimated ANN/SNN forward energy ratio is approximately 38.9× at T=2 and 2.8× at T=32. We emphasize that this is only a coarse compute-side reference, not a hardware-level energy claim.
>
> Finally, pure ANN-to-SNN conversion is not completely “free” either: it still requires a calibration forward pass to determine the voltage scaling / threshold normalization. In our SpikingJelly ANN-to-SNN conversion pipeline, this conversion-only stage takes about 245 s and only 0.32 GB peak GPU memory. As shown in Table 3, the SpikingJelly baseline yields only 10.00% at T=2 and 93.90% at T=32, whereas our method reaches 90.05% and 96.38%, respectively. By comparison, our 50-epoch post-conversion refinement takes 1003 s with a peak GPU memory usage of 2.36 GB. In other words, pure conversion is cheap but insufficient at low T, while our post-conversion refinement adds moderate extra training cost and closes most of the fidelity gap exactly where pure conversion fails.

---

> > ### Author Rebuttal · Reviewer_MC9t · 2026-04-04
> >
> > I appreciate the author’s rebuttal response; however, I will keep my original score because my main concerns remain insufficiently addressed.

---

> > > ### Author Response · Authors · 2026-04-05
> > >
> > > We sincerely thank you for the constructive feedback. We acknowledge that this paper does not include direct measurements of energy efficiency or deployment on neuromorphic hardware. Therefore, our efficiency discussion is limited to simulation-based results and a coarse compute-side energy proxy following prior work.

---

### Decision · Program_Chairs · 2026-04-30

**Decision:**

Accept (regular)

**Comment:**

The manuscript was reviewed by three reviewers. Following the rebuttal, all reviewers expressed positive overall assessments.

The reviewers found the submission well motivated, technically robust, practically meaningful in its methodology, and supported by promising results.

The Area Chair agrees with the reviewers’ consensus and recommends acceptance.

The authors should, still, carefully address the reviewers’ suggestions in the final version.

Congrats!